# Does Extraesophageal Reflux Support the Development of Lung Adenocarcinoma? Analysis of Pepsin in Bronchoalveolar Lavage in Non-Smoker Patients

**DOI:** 10.3390/cancers16152687

**Published:** 2024-07-28

**Authors:** Petra Zemanova, Michal Vocka, Zdislava Vanickova, Frantisek Liska, Ludmila Krizova, Josef Kalab, Jiri Votruba

**Affiliations:** 1First Clinic of Tuberculosis and Respiratory Diseases, First Faculty of Medicine, Charles University in Prague, General University Hospital in Prague, U Nemocnice 2, 12808 Prague, Czech Republic; josef.kalab@vfn.cz (J.K.); jiri.votruba@vfn.cz (J.V.); 2Department of Oncology, First Faculty of Medicine, Charles University in Prague, General University Hospital in Prague, U Nemocnice 2, Prague 2, 12808 Prague, Czech Republic; michal.vocka@vfn.cz (M.V.); ludmila.krizova@vfn.cz (L.K.); 3Institute of Medical Biochemistry and Laboratory Diagnostics, First Faculty of Medicine, Charles University in Prague, General University Hospital in Prague, U Nemocnice 2, 12808 Prague, Czech Republic; zdislava.vanickova@vfn.cz; 4Institute of Biology and Medical Genetics, Purkyne Institute, First Faculty of Medicine, Charles University in Prague, Albertov 4, 12800 Prague, Czech Republic; frantisek.liska@vfn.cz

**Keywords:** extraesophageal reflux, pepsin, lung adenocarcinoma, lung sarcoidosis

## Abstract

**Simple Summary:**

In our study, we investigated the association between pepsin levels in bronchoalveolar lavage and factors such as hemorrhagic/non-hemorrhagic character of BAL, type of lung disease, tumor location, patient demographics, and smoking history. Our objective was to investigate the possible influence of extraesophageal reflux on the development of lung adenocarcinoma. Contrary to our hypothesis, we did not find significant differences in pepsin levels among patients with lung adenocarcinoma, pulmonary metastases, and lung sarcoidosis. Pepsin levels were even lower in lung adenocarcinoma patients after excluding hemorrhagic samples. By rigorously excluding smokers and carefully classifying patients, we minimized biases common in similar studies. However, our study was limited by small sample sizes, especially in the sarcoidosis group, and the absence of standardized thresholds for BAL pepsin as a reflux biomarker.

**Abstract:**

The significance of extraesophageal reflux as a risk factor in lung adenocarcinoma has been understudied. In this study, we investigated whether extraesophageal reflux leads to higher pepsin concentrations in bronchoalveolar lavage (BAL) in patients with lung adenocarcinoma compared to controls. Subjects were recruited from non-smoker patients (lifelong non-smokers and ex-smokers with more than 5 years of non-smoking history) who had undergone bronchoscopy due to pulmonary abnormalities on a CT scan and met the inclusion criteria. Based on histological verification of the lung process, the patients were divided into three groups: (1) lung adenocarcinoma, (2) pulmonary metastases, and (3) lung sarcoidosis. Lung adenocarcinoma cases were further categorized as central or peripheral. BAL samples collected during bronchoscopy were quantitatively analyzed by enzyme-linked immunosorbent assay (ELISA) to measure pepsin levels. No statistically significant difference in pepsin concentration was observed between the lung adenocarcinoma group and control groups (*p* = 0.135). After excluding hemorrhagic BAL samples, the pepsin concentration was significantly the lowest in patients with lung adenocarcinoma (*p* = 0.023) compared to the control groups. The results of the study do not support the hypothesis of a higher occurrence of extraesophageal reflux (evaluated as the amount of pepsin in BAL) in non-smoker patients with lung adenocarcinoma.

## 1. Introduction

Extraesophageal reflux is defined as a leak of gastroesophageal fluid above the upper esophageal sphincter. The problem is associated with a wide spectrum of ENT (ear-nose-throat) and pulmonary diseases [1,2]. Pepsin, bile acids, bile acid salts, and pancreatic enzymes play particularly important roles in the pathogenesis of extraesophageal reflux. Acidity is no longer considered the primary pathogenetic factor during extraesophageal reflux episodes [3]. The basic characteristics of gastroesophageal reflux and extraesophageal reflux are summarized in Table 1.

There are eight isoforms of pepsin, differing in the optimal pH for their action. Pepsin reaches its maximum activity at pH values between 1.9 and 3.6, while at pH above 6.5, it becomes inactive, and at pH above 8, it is denatured. In the pH range of 6.5–8, pepsin is inactive but stable, with the ability to bind to mucosal cells and to reactivate upon reacidification. This can occur after it enters a cell or during a reflux episode [6]. This may explain the fact that even weakly acidic refluxate can damage the mucosa [7]. The role of pepsin in damaging the esophageal epithelium was experimentally demonstrated by Goldberg et al. as early as 1969 [8]. Two mechanisms for the pepsin-elicited damage to the mucosa have been described—breaching the protective mucosal barriers and a cytotoxic action upon entering cells. Pepsin inhibits the gene expression of protective mucosal proteins, such as mucins MUC 2, 3, 5AC, 5B [9]. The presence of pepsin in the laryngeal mucosa was correlated with the depletion of carbonic anhydrase isoenzyme III (CA III) in mucosal cells. CAIII catalyzes reactions leading to the production of bicarbonate and, thus, to the neutralization of acid reflux. In this way, CA III directly reduces pepsin activity [10]. The presence of pepsin in the laryngeal mucosa in patients with extraesophageal reflux is associated with the depletion of stress proteins Sep70 (squamous epithelial stress protein 70), Sep53 (squamous epithelial stress protein 53), and Hsp70 (heat shock protein 70), i.e., proteins the expression of which increases when a cell is exposed to a stress stimulus. This may indicate an impaired protection of the epithelial cell from the damage caused by the refluxate [11].

Pepsin is internalized by laryngeal epithelial cells through receptor-mediated endocytosis. Pepsin is transported by late endosomes or TRGs (trans-sacculotubular elements) into the cell. Moreover, through stimulation of expression of proinflammatory mediators and cytokines, pepsin contributes to chronic inflammation [12].

The above-mentioned components of the refluxate-induced tissue damage can be potentially tumorigenic. Indeed, the role of gastroesophageal reflux (GER) on the development of Barrett’s esophagus, a well-known precancerous lesion for esophageal adenocarcinoma, is indisputable [13].

Besides damage to the esophagus, reflux was suggested to play a role also in the development of laryngeal carcinoma [14,15,16,17,18]. However, determining the extent to which the reflux alone is causative in the development of these tumors is challenging. The increased prevalence of reflux in patients with laryngeal carcinoma might be explained by the concurent use of tobacco and/or alcohol products in the studied population. Smoking and alcohol consumption undoubtedly contribute to the development of laryngeal carcinoma as well as reduce the tone of the lower esophageal sphincter, thereby promoting the development of GER. Nevertheless, Bacciu and colleagues demonstrated an increased prevalence of reflux in a group of patients with laryngeal cancer who neither smoked nor consumed alcohol [15]. Reflux induces chronic mucosal irritation, leading to chronic inflammation, which can subsequently progress to dysplastic changes and invasive carcinoma. This hypothesis is supported by the fact that the laryngeal mucosa is much more sensitive to the irritative effects of reflux than the esophageal mucosa [19]. In contrast to laryngeal carcinoma, the association between extraesophageal reflux and the development of non-small cell lung cancer has received less research attention [20,21,22,23].

In our study, we investigated whether (i) extraesophageal reflux has a higher prevalence in patients with lung adenocarcinoma compared to controls and whether (ii) the determination of pepsin concentration in BAL using the ELISA method can be used to detect the presence of extraesophageal reflux in the lower respiratory tract.

## 2. Materials and Methods

This prospective case-control study recruited participants from patients referred for bronchoscopy at the First Department of Tuberculosis and Respiratory Diseases of the General University Hospital in Prague, Czech Republic, in 2019–2023 for diagnosis of pulmonary lesions.

The inclusion criteria were age ≥18 years, a pulmonary lesion requiring bronchoscopy diagnosis, histological verification of lung adenocarcinoma, metastatic lung involvement, or lung sarcoidosis. Additional criteria included lifelong or at least 5 years non-smoking, abstinence from alcohol, and no use of proton pump inhibitors and H2 blockers within the last month.

Patients were categorized into three groups based on the histological diagnosis of the pulmonary lesion: a group with newly diagnosed lung adenocarcinoma, a group with metastatic lung involvement from a primary extrapulmonary tumor, and a group with lung sarcoidosis. Based on CT or PET/CT imaging and/or by tumor location within 2 cm of the proximal bronchial tree according to bronchoscopy results, patients with lung adenocarcinoma were further categorized as those with central or peripheral tumors according to the contact with hilar structures (i.e., lobar bronchi, lobar or main pulmonary arteries, main pulmonary veins).

The smoking status was assessed through a pre-inclusion questionnaire and confirmed at a follow-up appointment. Only patients with consistent answers at both time points were included in the study.

### 2.1. Sample Collection and Processing

BAL samples were collected as a part of the bronchoscopic diagnostic algorithm of the pulmonary lesion. Briefly, a flexible bronchoscope was inserted into the airway via a laryngeal mask or endotracheal cannula. After insertion of the bronchoscope into the subsegmental bronchus of the right middle lobe, lavage with 100 mL of sterile physiological saline solution was performed. The aspirated BAL was stored on dry ice and laboratory-processed within one hour of collection.

Two 1 mL aliquots were centrifuged (3000 rpm for 15 min at 3 °C) and the obtained supernatants were stored at −70 °C until further processing. Pepsin concentration in the aliquots was measured using a commercially available Human Pepsin ELISA kit (BlueGene Biotech, Shanghai, China). Samples and buffer were incubated together with the PP-HRP conjugate in a pre-coated plate for one hour. After the incubation period and being-washed five times, samples were incubated with a substrate for HRP enzyme. Finally, a stop solution (tetramethylbenzidine) was added to stop the reaction. The color change was measured spectrophotometrically. The sensitivity of this assay is 1.0 pg/mL.

### 2.2. Statistics

Basic descriptive statistics were presented as mean ± SD or median (IQR). Wilcoxon test was used to compare two groups (e.g., lifelong non-smokers and former smokers); Kruskal–Wallis test was used for more groups (lung adenocarcinoma, lung metastases, lung sarcoidosis). χ^2^ test was used to assess the homogeneity of the groups regarding sex, smoking status, and presence of hemorrhagic BAL.

The association between pepsin concentration in BAL and independent covariates was analyzed using a linear regression model. All available covariates (age, sex, smoking status, and hemorrhagic character of the BAL sample) were included in the model. Due to a violation of the normality assumption, two subjects with outlying pepsin values (>1000) were excluded from the analysis. Diagnostic plots of the final linear regression model (Appendix A) show that there is no major violation of linear model assumptions. Detailed descriptions of the diagnostic plots are as follows:
-Residuals vs. Fitted Plot: The relationship between predictors and the outcome appears linear, indicating an appropriate fit of the linear model.-Normal Q-Q Plot: There is no significant deviation from normality in the residuals, as supported by the Shapiro–Wilk normality test (*p*-value = 0.644).-Scale-Location Plot: The residuals are randomly spread, suggesting that the assumption of homoscedasticity is likely not violated.-Residuals vs. Leverage Plot**: There are no influential observations that could unduly affect the model’s results.

All data analyses were performed using The R Project for Statistical Computing, version 4.2.2 [24].

## 3. Results

### 3.1. Patient Characteristics

Seventy-one patients were enrolled in the study and classified into three groups based on histological findings. Group I consisted of 30 patients with newly diagnosed non-small cell lung adenocarcinoma. There was a balanced sex distribution in this group, with a median age of 73 years. Group II included patients with histologically verified metastatic lung involvement of extrapulmonary origin, most commonly breast cancer (27.59%) and colorectal cancer (20.69%). The median age of this group was 68 years. The sex ratio was 0.81. Group III consisted of patients with a pulmonary form of sarcoidosis (median age of 55.5 years, sex ratio 1). Table 2 presents the basic characteristics of the patients in these groups, Appendix A provide detailed information of each group.

### 3.2. BAL Pepsin Concentration

BAL pepsin concentrations among the three groups were compared using the Kruskal–Wallis test. The median pepsin concentrations were 390.5 pg/mL in Group I, 449.9 pg/mL in Group II, and 447.5 pg/mL in Group III. Although no statistically significant difference in pepsin concentrations was observed, there was a trend contrary to the hypothesized expectation, i.e., slightly higher pepsin concentrations were detected in metastatic and sarcoidosis groups (*p* = 0.135, Kruskal–Wallis test; Figure 1, left). No association between age and BAL pepsin concentration was detected (Appendix A).

Following the exclusion of the hemorrhagic samples, pepsin concentration was significantly lowest in the Group I (*p* = 0.023, Kruskal–Wallis test). The median pepsin concentrations were then 350.9 pg/mL in Group I, 439.45 pg/mL in Group II, and 503.8 pg/mL in Group III. No statistically significant difference in pepsin concentrations between Group II and III was demonstrated (*p* = 0.371, Wilcoxon test; Figure 1, right).

The linear regression model evaluating the associations between age, sex, smoking status, and hemorrhagic character of BALs as the independent variables and BAL pepsin concentrations as the dependent variable identified the smoking status (former smokers) and hemorrhagic character of BALs to be the significant predictors of BAL pepsin concentrations (Table 3). According to the linear regression model, pepsin concentrations in the BAL did not differ between the groups with peripherally and centrally located lung adenocarcinoma (Figure 2). BAL pepsin concentrations were compared between patients with and without a history of smoking, revealing significantly higher (*p* = 0.001) pepsin concentrations in former smokers (median 458 pg/mL; IQR 399–546) than in life-long non-smokers (median 351 pg/mL; IQR 257–447) (Figure 3, Table 4). Two samples in our study showed exceptionally high pepsin concentration above 1000 pg/mL, both originating from former smokers. The comparison of pepsin concentrations between hemorrhagic (median 484 pg/mL, IQR 404–581) and non-hemorrhagic (median 395 pg/mL, IQR 280–482) BAL samples showed significantly higher levels of pepsin (*p* = 0.01) in hemorrhagic BALs (Figure 4, Table 4).

## 4. Discussion

In our study, we investigated the associations between BAL pepsin concentration and various factors such as non/hemorrhagic character of BAL, tumor location, age, and sex. Additionally, we examined the association between BAL pepsin concentrations and the type of the pulmonary findings (primary lung adenocarcinoma, lung metastases, lung sarcoidosis) to evaluate the hypothesis that extraesophageal reflux contributes to the development of lung adenocarcinoma, which is relatively more common in non-smokers than other types of lung carcinoma.

It was expected that should there be a causal association between extraesophageal reflux and the development of lung adenocarcinoma, and that we would observe a higher occurrence of extraesophageal reflux (evaluated as the amount of pepsin in the BAL samples) in patients with this type of tumor than in patients with other lung affections, whether malignant (metastases from tumors originating elsewhere) or non-malignant (lung sarcoidosis). However, our study results do not support this hypothesis as no statistically significant difference in pepsin concentrations was found among patients with lung adenocarcinoma, pulmonary metastases, and lung sarcoidosis. After excluding hemorrhagic BAL samples, the BAL pepsin concentration in patients with lung adenocarcinoma was even significantly lower than in the control groups.

Our study’s main strength lies in its rigorous differentiation between lifelong non-smokers and former smokers and in the exclusion of current smokers from the study. Smoking is a known cofactor of gastroesophageal reflux, contributing to lower esophageal sphincter tone, altered esophageal motility, delayed stomach emptying, increased gastric acid secretion, and reduced saliva production, which normally neutralizes reflux acidity [25]. Importantly, smoking is also a well-established risk factor for lung carcinoma, with smokers facing up to a 20-fold higher risk compared to non-smokers [26]. This strong association between smoking and both reflux and lung carcinoma underscores the importance of checking for smoking status when investigating the potential link between gastroesophageal reflux disease (GERD) and lung cancer.

Excluding smokers from our study mitigated a significant potential bias present in many studies examining the association between extraesophageal reflux and lung carcinoma. For instance, Hsu et al. reported an increased lung cancer risk in patients with GERD compared to those without GERD in an epidemiological study, but the inclusion of smokers was a notable limitation [20]. Similarly, Lin Li et al. used Mendelian randomization analysis to suggest GERD increased the risk of NSCLC, but did not specify smoking status [21].

Vereczkei et al. performed a cohort study with 50 patients of resectable non-small cell lung cancer. In their study, GERD was evaluated using 24-h pH-metry. Reflux was detected in 40% of patients with lung adenocarcinoma and 32% of those with squamous cell lung carcinoma, respectively. The original hypothesis, which expected a significantly higher reflux frequency among patients with lung adenocarcinoma, was not confirmed. It is, however, necessary to point out that 40% of patients with lung adenocarcinoma and 52% squamous lung carcinoma in their study were smokers. This could also explain the high rate of reflux disease in their study group, which is higher than the population prevalence reported in the literature [22]. In a retrospective case study Amernath et al. demonstrated a higher prevalence of GERD in patients with lung adenocarcinoma without a history of smoking compared to a control group without a diagnosis of non-small cell lung cancer (20.4% vs. 11.6%, *p* < 0.001). This suggests that GERD could be a risk factor for lung cancer even among nonsmokers (OR = 1.86, CI = 1.26–2.73) [23]. However, the study has notable limitations. Despite including a relatively large cohort of 543 participants and focusing on non-smokers, the reliability of both the GERD diagnosis and non-smoking status is questionable. The study included GERD patients identified only based on their medical records, i.e., international classification of diseases (ICD) code for GERD in a database without verification either by gastrofibroscopy with biopsy or by 24-h pH-metry. Additionally, the non-smoking status was determined only on the basis of the absence of a smoking record in the patient history, which, as the authors admit, might have led to the inclusion of an unknown proportion of smokers whose patient records were incompletely filled in.

The fact that the smoking status in our study was determined only on the basis of direct questioning and not on the basis of an objective method (e.g., cotinine detection) can be also considered a limitation of our study. However, as cotinine detection would provide relevant results only several weeks after smoking cessation and our study distinguished between lifelong non-smokers and ex-smokers with more than 5 years of non-smoking history, this would likely not add much to the classification of our patients. Given the long duration since smoking cessation, it is somewhat unexpected that BAL pepsin concentrations remain significantly higher in ex-smokers compared to lifelong non-smokers.

The highest pepsin concentration in BAL samples was detected in the group with lung sarcoidosis, although the finding did not reach statistical significance. However, our study’s limitation is the small size of the lung sarcoidosis group, which consisted of only 12 participants. Sarcoidosis patients were chosen as the control group because there is currently no known mechanism linking GERD and sarcoidosis. Further studies with a larger cohort are needed to investigate the hypothesis of elevated pepsin concentration in BAL among patients with pulmonary sarcoidosis.

Fukui et al. proposed that peripherally located lung adenocarcinoma probably arises from type II pneumocytes and Clara cells, whereas centrally located lung adenocarcinoma likely originates from glandular epithelium [27]. This epithelium being exposed to refluxate on the surface layer is more susceptible to its effects. However, the study did not find evidence supporting a higher incidence of reflux in patients with centrally located lung adenocarcinoma compared to those with peripherally located lung adenocarcinoma.

In our study, we confirmed higher BAL pepsin concentrations in hemorrhagic samples compared to in non-hemorrhagic samples (*p* = 0.01). False positivity in pepsin detection in hemorrhagic samples using the ELISA can be caused by interference of hemoglobin and other blood proteins with the reagents used in the ELISA test [28]. For this reason, hemorrhagic samples were excluded.

Since pepsin is normally not present at high concentrations in the lower airways, the assumption that its determination in BAL could serve as a biomarker of aspiration is reasonable. It can be argued that, unlike questionnaires, endoscopy with biopsy, pH-metry and multichannel intraluminal impedance, the use of BAL pepsin concentration to detect extraesophageal reflux in the lower airways is not an established and standardized method. Unfortunately, questionnaires such as RSI (reflux symptom index) could be used in our patients to significant overlap in symptoms between extraesophageal reflux and lung disease (e.g., cough, dyspnea, hoarseness, chest pain, etc.). The other methods mentioned above would mean subjecting the patients to an additional invasive examination. While pH-metry is effective in detecting acid reflux, it cannot distinguish between acid and non-acid reflux, which can be clinically relevant. Non-acid reflux, which contains bile acids, pancreatic enzymes, and pepsin, is particularly implicated in the pathogenesis of pulmonary diseases. Prolonged exposure to these substances can trigger an inflammatory reaction and leads to airway remodeling and damage to lung tissue damage. Multichannel intraluminal impedance pH monitoring (MII-pH monitoring) is a diagnostic method that provides information on both acid and non-acid reflux. Despite its advantages, such as the ability to detect non-acid reflux, this method has its drawbacks, including technical complexity, patient discomfort, and interpretation challenges [4,5].

Collecting BAL during bronchoscopy, which is already part of diagnostic procedures for these patients, is much more convenient. On the other hand, we acknowledge that this limits our selection of the control group to patients with existing lung disease who are already planned for bronchoscopy. Recruiting volunteers from an adjusted general population would be ethically doubtful due to the invasiveness of bronchoscopy and the relatively older age of potential participants.

Unlike the esophageal mucosa, the airway mucosa lacks defensive mechanisms against reflux fluid components. In fact, it is more susceptible to reflux-induced damage. However, a general statement that any micro-aspiration of reflux into the airways leading to the detection of pepsin in BAL would be causally associated with the development or worsening of lung disease is not justified. Therefore, it is crucial to determine the pathological threshold of pepsin concentration and understand the factors that influence elevated pepsin levels in BAL before implementing this method in routine clinical practice. Currently, there is no established cut-off value for BAL pepsin concentration that definitively indicates extraesophageal reflux. Literature suggests that the pepsin level may be influenced by factors other than aspiration (such as the contamination of the bronchoscope with contents of the upper airways, the time from the last reflux episode at the time of performing BAL, or the patient position) [29].

In the future, it will be crucial to undertake further studies specifically aimed at establishing clinically relevant cut-off values for pepsin concentration in BAL. These studies should encompass larger and more diverse cohorts, incorporating participation from healthy volunteers. Additionally, defining all factors that influence pepsin levels in BAL will be essential.

The exact mechanism by which reflux may contribute to the development of lung adenocarcinoma remains unclear. Our study did not focus on the causal pathways linking reflux to lung cancer tumorigenesis. Based on research on the impact of reflux on esophageal adenocarcinoma, we can hypothesize that chronic inflammation caused by reflux may significantly contribute to increased cellular and tumor transformation in pathogenesis of lung cancer. Validating this theory though a single-cell transcriptomics study would be appropriate. This approach could provide valuable insights into how inflammation triggered by reflux influences the cellular and molecular processes involved in lung adenocarcinoma development. By examining the gene expression profiles of individual cells, we may better understand the heterogeneous responses to reflux and pinpoint the critical mechanisms underlying cancer development.

The limitations of our study include the small sample size of groups and methodological considerations. The limited number of participants in the lung sarcoidosis group, with only 12 participants, may have affected the statistical power of our comparisons. Additionally, the use of BAL pepsin concentration as a biomarker for extraesophageal reflux in the lower airways is not standardized. While less invasive than pH-metry, it lacks established cut-off values and may be influenced by various factors, such as the presence of hemorrhagic BAL samples which can lead to false positive results.

## 5. Conclusions

We performed a study comparing the presence of pepsin in BAL contents in currently non-smoking patients with lung adenocarcinoma (representing a primary lung tumor possibly associated with GERD), metastatic tumors (representing tumors likely not associated with GERD), and lung sarcoidosis (a non-malignant process). We did not find any differences in BAL pepsin concentrations among these three groups. After excluding hemorrhagic BAL samples, the group of patients with lung adenocarcinoma exhibited the significantly lowest pepsin concentration.

## Figures and Tables

**Figure 1 cancers-16-02687-f001:**
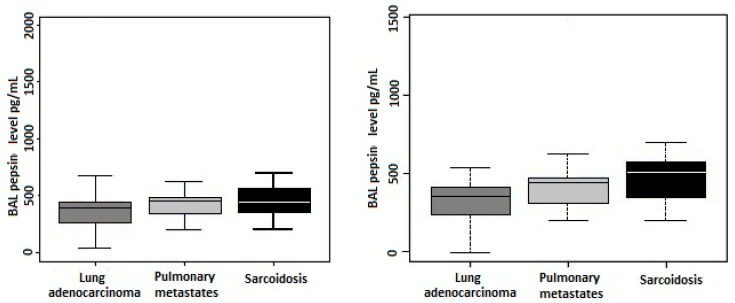
Comparison of BAL pepsin concentrations between groups (**left**) including hemorrhagic BAL samples and (**right**) excluding these samples. In the analysis including hemorrhagic samples, no statistically significant difference was detected overall (*p* = 0.135, Kruskal–Wallis test). However, there was a trend towards higher pepsin concentrations in the metastatic and sarcoidosis groups compared to the lung adenocarcinoma group. Upon excluding hemorrhagic samples (**right**), pepsin concentration was lowest in the group of patients with lung adenocarcinoma (*p* = 0.023, Kruskal–Wallis test). The boxplots illustrate the median (central line), interquartile range (box), and minimum–maximum range (whiskers), with outliers represented by circles.

**Figure 2 cancers-16-02687-f002:**
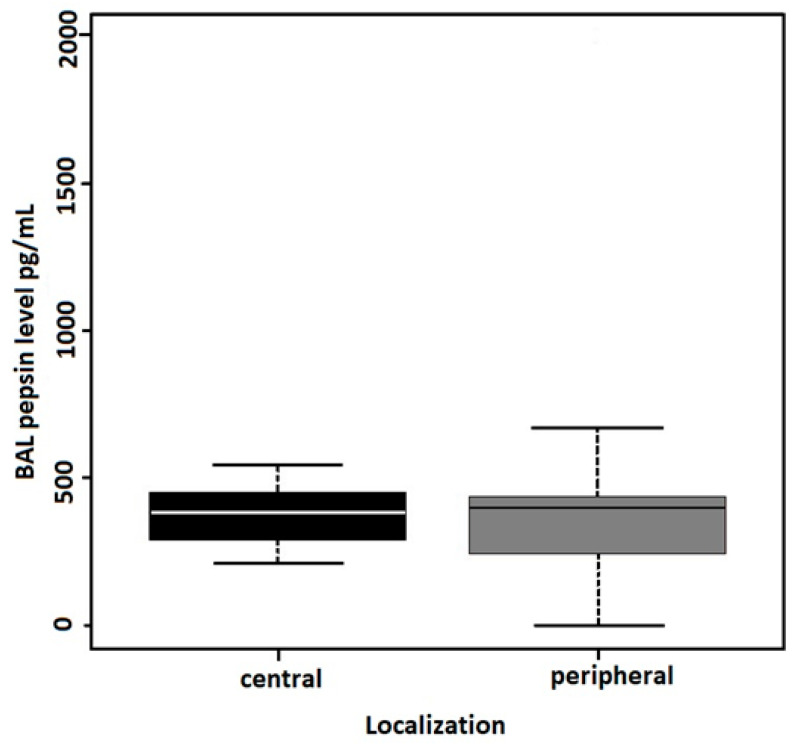
BAL pepsin concentration in peripheral and central lung adenocarcinoma. No statistically significant difference was detected (*p* = 0.667, Wilcoxon test).

**Figure 3 cancers-16-02687-f003:**
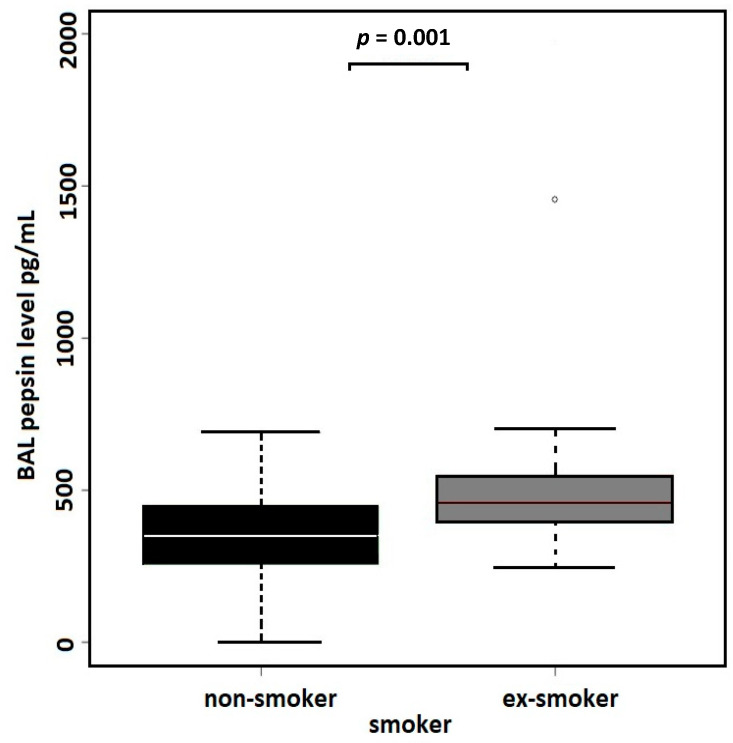
BAL pepsin concentration in the group of lifelong non-smokers versus former smokers (*p* = 0.001, Wilcoxon test). Outliers represented by circles.

**Figure 4 cancers-16-02687-f004:**
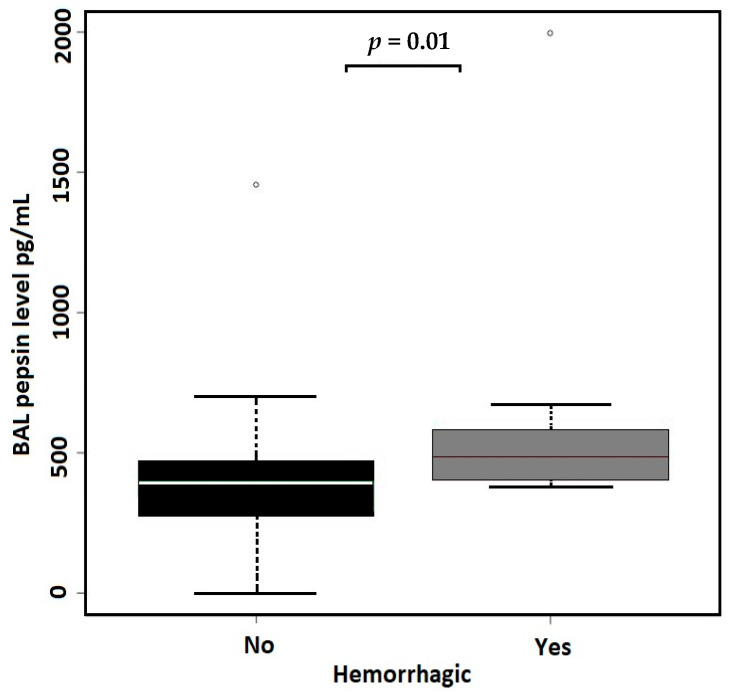
BAL pepsin concentration in hemorrhagic versus non-hemorrhagic BAL samples (*p* = 0.01, Wilcoxon test). Outliers represented by circles.

**Table 1 cancers-16-02687-t001:** Shows the basic characteristics of gastroesophageal reflux and extraesophageal reflux [4,5].

	Gastroesophageal Reflux	Extraesophageal Reflux
Definition	Occurs due to the backflow of gastric contents into the esophagus. If DeMeester criteria are met (pH < 4 or > 7 in more than 50 episodes within 24 h, or reflux lasting longer than 1 h within 24 h according to pH monitoring), it is considered pathological reflux.	Occurs when refluxate rises above the upper esophageal sphincter.
Symptoms	Heartburn, regurgitation, dysphagia, chest pain	Chronic cough, postnasal drip syndrome, hoarseness or sore throat, halitosis, dental erosions, globus sensation, chest pain, dyspnoea, sleep disturbance, wheeze
Diagnostic tools	Medical history, physical examination, esophagogastroduodenoscopy, 24-h pH-metry, 24-h multichannel intraluminal impedance (MII), esophagogram, esophageal manometry Trial of medication: response to acid-suppressing medications (proton pump inhibitors or PPIs, H2-receptor antagonists) Questionnaires: various standardized questionnaires—such as GERD-Q	Medical history, physical examination, fibrolaryngoscopy, 24-h pH-metry, 24-h multichannel intraluminal impedance (MII), questionnaires: such as RSI
Significance of acidity	Yes	Low

**Table 2 cancers-16-02687-t002:** Basic characteristics of the groups.

Group	Lung Adenocarcinoma	Pulmonary Metastases	Lung Sarcoidosis	
	(Group I)	(Group II)	(Group III)	*p*-value
Group size (%)	30/71 (42.25%)	29/71 (40.85%)	12/71 (16.9%)	-
Age (Median [IQR])				
	73 (69, 76)	68 (62, 76)	56 (45, 59)	<0.001 ^a^
Sex				
Male	15/30 (50%)	13/29 (44.8%)	6/12 (50%)	0.912 ^b^
Female	15/30 (50%)	16/29 (55.2%)	6/12 (50%)	
Smoking status				
Never	15/30 (50%)	16/29 (55.2%)	8/12 (66.7%)	0.618 ^b^
Former	15/30 (50%)	13/29 (44.8%)	4/12 (33.3%)	
Hemorrhagic BAL				
No	23/30 (76.7%)	24/29 (82.8%)	11/12 (91.7%)	0.515 ^b^
Yes	7/30 (23.3%)	5/29 (17.2%)	1/12 (8.3%)	

(^a^) Kruskal–Wallis test. (^b^) Chi-square contingency table test.

**Table 3 cancers-16-02687-t003:** Linear regression model for prediction of pepsin concentrations in BAL.

	Estimate	95% Confidence Interval for Coefficient		*p*-Value
		Low	High	
(Intercept)	474.957	322.4074	627.5075	4.44 × 10^−8^
Lung adenocarcinoma	−75.623	−138.457	−12.7887	0.019119
Age	−1.777	−4.24291	0.688678	0.154757
Female sex	16.628	−42.0543	75.3099	0.573244
Former smoker	107.88	49.06649	166.6941	0.000508
Hemorrhagic BAL	99.924	23.23115	176.6161	0.011489

Adjusted R^2^ = 0.2762.

**Table 4 cancers-16-02687-t004:** Shows differences in BAL pepsin concentrations: (**A**) Between hemorrhagic and non-hemorrhagic samples. (**B**) Between lifelong non-smokers and former smokers across all samples. (**C**) Between lifelong non-smokers and former smokers after excluding hemorrhagic samples. (**D**) Between peripherally and centrally located lung adenocarcinoma.

**(A)** **All Samples, N = 71**		**Non-Hemorrhagic BAL**	**Hemorrhagic BAL**	**Effect Size ^b^**	***p*-Value**
BAL PEPSIN levels		N = 58	N = 13		
	Median (MAD) [I.Q.]	395.35 (133.8) [280.2, 472.43]	483.9 (139.4) [404, 581]	0.657	0.01 ^a^
**(B)** **All Samples, N = 71**		**Non-Smokers**	**Former Smokers**		
BAL PEPSIN levels		N = 39	N = 32		
	Median (MAD) [I.Q.]	350.9 (143.8) [257.45, 446.6]	458.4 (98.7) [399.17, 545.95]	0.856	0.001 ^a^
**(C)** **Non-Hemorrhagic Samples, N = 58**		**Non-Smokers**	**Former Smokers**		
BAL PEPSIN levels		N = 34	N = 24		
	Median (MAD) [I.Q.]	329.5 (145.7) [246.85, 440.42]	451.5 (82.5) [392.18, 487.3]	0.986	0.002 ^a^
**(D)** **Lung Adenocarcinoma,** **N = 30**	**Central**		**Peripheral**		
BAL PEPSIN levels		N = 11	N = 19		
	Median (MAD) [I.Q.]	382.9 (138.6) [288.45, 450.15]	398.2 (176.7) [243.8, 434.9]	0.093	0.667 ^a^

(^a^) Wilcoxon test. (^b^) standardized difference in median.

## Data Availability

Data available on request due to privacy/ethical restrictions.

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
