# Peer review of "Does Extraesophageal Reflux Support the Development of Lung Adenocarcinoma? Analysis of Pepsin in Bronchoalveolar Lavage in Non-Smoker Patients"

_cancers, 2024, doi:10.3390/cancers16152687_

Round 1

Reviewer 1 Report

Comments and Suggestions for Authors

The manuscript evaluated extraesophageal reflux as a risk factor in lung adenocarcinoma via evaluating whether extraesophageal reflux leads to a higher pepsin 18 concentration in bronchoalveolar lavage (BAL) in patients with lung adenocarcinoma compared to two controls (patients with pulmonary metastasis or with lung sarcoidosis). Patients had no smoking history or had stopped smoking for at least five years. No difference was found among patients with adenocarcinoma versus the control groups. And contrary to expectation after excluding hemorrhage the pepsin concentration was actually lower in patients with lung and no carcinoma. Findings suggest that extraesophagealreflux using pepsin as a surrogate via bronchial religion lavage is not associated with Lung adenocarcinoma in non-smokers.

The first assumption in this paper is that pepsin levels correlate with the amount of extraesophageal reflux. Indeed it is support this is weak. Other surrogates or direct measurements reflux such as via it's official pH measurementsor other means or correlation with the severity of reflux are other potential surrogate markers that could be assessed. The authors recognize this but chose not to use these other methods either symptoms can mimic Barca genic carcinoma

The number of patients in each group is small leading to the suggestion that there may not be enough power to come to the conclusion of no difference. Of note, the P value was 0.135 thus was approaching significance. No power calculation is provided.

Minor:

There are several spelling errors.

Comments on the Quality of English Language

English quality is ok, but there were several spelling errors. 

Author Response

Dear Reviewer,

Thank you for your comprehensive evaluation of our manuscript.

 We understand that our findings may not fully validate the hypothesis regarding a higher prevalence of extraesophageal reflux in non-smoking patients with lung adenocarcinoma. This is a pilot study and should be considered as offering preliminary findings that require further validation. We agree that future studies should aim for larger and more balanced sample sizes across all groups to enhance the validity and generalizability of the results.

The statistical analysis has been described in more detail, and diagnostic plots of the linear regression model have been added (Figures S1-S4 in the supplement), which show that there are no major violations of the linear model assumptions. Confidence intervals and effect sizes have also been included.

Thank you once again for your valuable feedback. We appreciate your time and effort in reviewing our manuscript.

Best regards

Petra Zemanová

Reviewer 2 Report

Comments and Suggestions for Authors

This study showed that BAL fluid pepsin content in lung cancer was not much different from  BAL fluid pepsin content in cancers metastatic to lung. But the authors did show an interesting lower BAL fluid pepsin content in non-hemorragic lung cancer compared to sarcoidosis or lung metastatic BAL fluid pepsin content. That is an interesting finding. Also of interest is the somewhat tangential finding that ex-smokers had higher BAL fluid pepsin content compared to never smokers.

English use is good, easily read. A few minor typos need correcting.

Line 39. Isn’t plain acid, H+, HCl a major irritant too ?

Line 51. Unclear where CAIII is depleted. In mucosa cells of…? Or where ? Do we know why CAIII depletion happens ? What about the other 10+ isoforma of CA ? Can they cross cover ?

Line 61. What is the “pepsin receptor”? Don’t we know pepsin is endocytosed yes, but that does not mean that pepsin has aa cognate receptor. Does it ?

Since gastoesophageal reflux is better known, and you do compare it with extraesophageal reflux, would a Table listing similarities and differences between the two be helpful to readers ? 

Line 179. A missing word and a missing letter. “Two samples in our study showed extremely pepsin concentratin above 1000 pg/mL, both originating from former smokers. “ Presumably “extremely” and “concentration” ?

Lines 207 to 210. The might be a problem with this statement: “However, the results of our study do not support this hypothesis as no statistically significant difference in pepsin concentrations was found among the groups. Suprisingly, after excluding hemorrhagic BAL samples, pepsin concentration was even significantly lower in the group of patients with lung adenocarcinoma than in the control groups.”   You did find a significant difference between the groups. Non-hemorragic BAL from lung cancer had significantly lower pepsin than metastatic cancer to lungs or sarcoidosis. Right ? Also did you not show significantly different BAL pepsin levels in ex-smokers compared to never smokers ?

Line 236. Do we know the corresponding values from pH metry in people with no GERD signs or symptoms ?

This study would have been much more informative if 24 hr pH metry had been performed prior to bronchoscopy. Also nice would have been measuring other gastric fluid markers - lipase, intrinsic factor, pH. Would measuring pepsinogen as well as pepsin have helped us understand what is happening better ? Did your pepsin measuring method cross-react with pepsinogen ?

Line 264. Error. “...GERD and sarcoidosis….” intended.

Comments on the Quality of English Language

This study showed that BAL fluid pepsin content in lung cancer was not much different from  BAL fluid pepsin content in cancers metastatic to lung. But the authors did show an interesting lower BAL fluid pepsin content in non-hemorragic lung cancer compared to sarcoidosis or lung metastatic BAL fluid pepsin content. That is an interesting finding. Also of interest is the somewhat tangential finding that ex-smokers had higher BAL fluid pepsin content compared to never smokers.

English use is good, easily read. A few minor typos need correcting.

Line 39. Isn’t plain acid, H+, HCl a major irritant too ?

Line 51. Unclear where CAIII is depleted. In mucosa cells of…? Or where ? Do we know why CAIII depletion happens ? What about the other 10+ isoforma of CA ? Can they cross cover ?

Line 61. What is the “pepsin receptor”? Don’t we know pepsin is endocytosed yes, but that does not mean that pepsin has aa cognate receptor. Does it ?

Since gastoesophageal reflux is better known, and you do compare it with extraesophageal reflux, would a Table listing similarities and differences between the two be helpful to readers ? 

Line 179. A missing word and a missing letter. “Two samples in our study showed extremely pepsin concentratin above 1000 pg/mL, both originating from former smokers. “ Presumably “extremely” and “concentration” ?

Lines 207 to 210. The might be a problem with this statement: “However, the results of our study do not support this hypothesis as no statistically significant difference in pepsin concentrations was found among the groups. Suprisingly, after excluding hemorrhagic BAL samples, pepsin concentration was even significantly lower in the group of patients with lung adenocarcinoma than in the control groups.”   You did find a significant difference between the groups. Non-hemorragic BAL from lung cancer had significantly lower pepsin than metastatic cancer to lungs or sarcoidosis. Right ? Also did you not show significantly different BAL pepsin levels in ex-smokers compared to never smokers ?

Line 236. Do we know the corresponding values from pH metry in people with no GERD signs or symptoms ?

This study would have been much more informative if 24 hr pH metry had been performed prior to bronchoscopy. Also nice would have been measuring other gastric fluid markers - lipase, intrinsic factor, pH. Would measuring pepsinogen as well as pepsin have helped us understand what is happening better ? Did your pepsin measuring method cross-react with pepsinogen ?

Line 264. Error. “...GERD and sarcoidosis….” intended.

Author Response

Dear Reviewer,

Thank you for your comprehensive evaluation of our manuscript.

Acidity is no longer regarded as the primary pathogenetic factor in extraesophageal reflux episodes. Non-acid reflux, which contains bile acids, pancreatic enzymes, and pepsin, plays a significant role in the pathogenesis of pulmonary diseases. Prolonged exposure to these substances can trigger inflammatory reactions, leading to airway remodelling and damage to lung tissue.

CAIII depletion typically refers to its reduced often in mucosal cells such as those in the oesophagus or other tissues affected by conditions like reflux or inflammation. The exact mechanisms causing CAIII depletion can vary but may involve factors like oxidative stress, inflammatory cytokines, or cellular damage.

You are correct; there is no known specific receptor for pepsin. Pepsin is indeed endocytosed by cells, but this process does not necessarily involve a dedicated receptor. Instead, pepsin is internalized through general endocytic pathways that do not require a specific pepsin receptor.

Thank you for your recommendations. The table comparing gastroesophageal and extraesophageal reflux has been added.

Clarified between which groups a non-significant difference in BAL pepsin concentrations was observed.

pH-metry is effective in detecting acid reflux. Non-acid reflux is particularly implicated in the pathogenesis of pulmonary diseases. Due to these reasons, We do not consider pH-metry for assessing reflux in the lower airways to be appropriate.

We also considered measuring bile acids; however, the high costs associated with their analysis using tandem mass spectrometry exceeded the financial resources of our study, leading us to forgo this measurement. In the future, measuring lipase could be considered. Like bile acids, lipase also indicates duodenogastric reflux, and it can also be measured using the ELISA method, similarly to pepsin. Testing with ELISA pepsin kit for pepsinogen was conducted prior to sample analysis, ensuring the absence of cross-reactivity.

Thank you once again for your valuable feedback. We appreciate your time and effort in reviewing our manuscript.

Best regards

Petra Zemanová

Reviewer 3 Report

Comments and Suggestions for Authors

Dear Editor,

I was pleased to review the manuscript entitled ''Does Extraesophageal Reflux Support the Development of Lung Adenocarcinoma? Analysis of Pepsin in Bronchoalveolar Lavage in Non-Smoker Patients''. The authors sought to investigate whether extraesophageal reflux leads to higher pepsin concentration in bronchoalveolar lavage (BAL) in patients with lung adenocarcinoma compared to controls. Based on histologic confirmation of the lung process, patients were divided into three groups, including (1) lung adenocarcinoma and two control groups with (2) pulmonary metastases or (3) pulmonary sarcoidosis. The authors did not observe a statistically significant difference in pepsin concentration between the lung adenocarcinoma group and the control groups. The study did not support the hypothesis of a higher occurrence of extraesophageal reflux in non-smoking patients with lung adenocarcinoma. I congratulate the authors for this comprehensive work. However, I think that the study will not contribute to the literature.  

Sincerely

Comments on the Quality of English Language

Moderate editing of English language required

Author Response

Dear Editor,

Thank you for your thoughtful review of our manuscript. We understand that our findings may not fully validate the hypothesis regarding a higher prevalence of extraesophageal reflux in non-smoking patients with lung adenocarcinoma. Nevertheless, we believe our study provides valuable insights into the role of pepsin in bronchoalveolar lavage as a potential marker for extraesophageal reflux and its relationship with lung adenocarcinoma. As a pilot study, it establishes important groundwork for future research and highlights the need for more extensive studies involving larger and more diverse populations.

Best regards
Petra Zemanová

Reviewer 4 Report

Comments and Suggestions for Authors

The authors present a manuscript in which they assessed whether lung tumors were linked to a potential gastroesophageal reflux based on the detection of pepsin. They conclude that association between gastroesophageal reflux and lung cancers might be associated to the smoking status.

1. The major issue with the report is that it might have used the wrong target enzyme. The subjects had been anesthesized and their airway was instrumented, which indicates that these patients had been fastened before. We do however know that pepsin concentrations are poor predictors of reflux episodes and this is especially the case after fasting (DOI: 10.3390/biomedicines12020398). Consequently, the conclusions are overstated with regard to that aspect and also with regard to the smoking status as this is pure speculation. The authors might do that in the discussion, but not in the conclusion.

2. Figure 1 is of too low quality to be readable. In addition, please adjust the figures accordingly so that they are accessible for color-blind readers. Moreover, a trend has not been investigated, so please don't mention one.

3. The regression model is insufficiently documented and reported. Which regression has been used? How has the model been validated? Do the data fulfill the regressions requirements? How have the predictors been chosen? What are the model diagnostics? The sample is small, so there might be overfitting.

4. Please use correct termini technici, it's not significant, but statistically significant.

5. Please cite R correctly by referencing the R core team and all packages that were used.

6. The discussion needs to be more focused on what has acutally been measured and not discuss extensively the aspect of smoking that has only been peripherally investigated.

7. Relevant data is missing such as an assessment of reflux symptoms using validated scores that might easily explain the seen differences.

8. Please discuss the limitations of the study appropriately.

Author Response

Dear Reviewer,

Thank you for your comprehensive evaluation of our manuscript.

I understand your concerns regarding the reliability of using salivary pepsin levels as a marker for extraesophageal reflux, especially given the semiquantitative approach of the Peptest method. This led us to explore whether a quantitative measurement of pepsin using ELISA in the lower airways could provide a more accurate assessment of extraesophageal reflux. Several studies have investigated the impact of reflux on exacerbating interstitial lung diseases and have utilized ELISA to measure pepsin in the lower airways as a more precise indicator of reflux.

Thank you for your recommendations. The statistical analysis has been described in more detail, and diagnostic plots of the linear regression model have been added (Figures S1-S4 in the supplement), which show that there are no major violations of the linear model assumptions. Confidence intervals and effect sizes have also been included.

Thank you for your constructive comments. We have enhanced the quality of the graphs, used the correct terminology, and accurately cited R by referencing the R Core Team. the discussion has been revised to focus more precisely on the measurements and results obtained in our study, including the limitations of our research.

Unfortunately, questionnaires such as the Reflux Symptom Index (RSI) could not be effectively used in our patients due to significant symptom overlap between extraesophageal reflux and lung disease (e.g., cough, dyspnoea, hoarseness, chest pain, etc.). Initially, we collected data using RSI questionnaires, but the observed overlap led us to discontinue their use due to their limited informativeness.

Thank you once again for your valuable feedback. We appreciate your time and effort in reviewing our manuscript.

Best regards

Petra Zemanová

Reviewer 5 Report

Comments and Suggestions for Authors

General comments

This study investigates the potential role of extraesophageal reflux (EER) in the development of lung adenocarcinoma by analyzing pepsin concentrations in bronchoalveolar lavage (BAL) samples from non-smoker patients. The authors compared pepsin levels among three groups: patients with lung adenocarcinoma, those with pulmonary metastases, and those with lung sarcoidosis. Despite the hypothesis that higher pepsin levels in BAL would indicate a link between EER and lung adenocarcinoma, the study found no statistically significant difference in pepsin concentrations between the groups. Interestingly, after excluding hemorrhagic samples, pepsin concentrations were significantly lower in the lung adenocarcinoma group compared to controls. The study highlights the importance of differentiating between non-smokers and former smokers, considering smoking as a confounding factor. The findings suggest that the previously reported association between EER and lung adenocarcinoma might be influenced by smoking history rather than a direct causal relationship.

While the study provides valuable insights into the potential role of EER in lung adenocarcinoma, it has several flaws and loopholes that need addressing to strengthen the validity and clinical relevance of its findings. Future research should aim to address these issues by expanding sample sizes, controlling for confounding factors, providing mechanistic insights, and ensuring consistent and detailed statistical reporting.

Research Loopholes and Flaws:

1.    Sample Size and Composition

      Flaw: The study includes a relatively small sample size, particularly in the sarcoidosis group (12 patients), which can limit the statistical power and reliability of the findings.

      Recommendation: Future studies should aim for larger and more balanced sample sizes across all groups to enhance the validity and generalizability of the results.

2.    Exclusion of Hemorrhagic BAL Samples:

      Flaw: The exclusion of hemorrhagic BAL samples is mentioned, but the rationale and potential impact on the results are not thoroughly explained. This exclusion might introduce bias or affect the interpretation of pepsin levels.

      Recommendation: Provide a clear justification for excluding hemorrhagic samples and include sensitivity analyses to show how their exclusion influences the results.

3.    Control for Confounding Factors:

      Flaw: While the study attempts to control for smoking status, other potential confounding factors such as diet, medication use, or other respiratory conditions that might affect pepsin levels are not adequately addressed.

      Recommendation: Incorporate a more comprehensive assessment of potential confounders and their control within the study design to ensure more accurate results.

4.    Lack of Mechanistic Insights:

      Flaw: The study does not delve into the mechanisms through which extraesophageal reflux might contribute to lung adenocarcinoma development. Understanding the biological pathways is crucial for establishing a causal relationship.

      Recommendation: Include mechanistic studies or propose potential biological pathways based on existing literature to strengthen the plausibility of the hypothesized link between EER and lung adenocarcinoma.

5.    Statistical Analysis Details:

      Flaw: The statistical analyses lack detailed reporting, such as confidence intervals and effect sizes. Additionally, the rationale for the choice of statistical tests is not fully explained, and there is no mention of multiple comparison corrections.

      Recommendation: Report detailed statistical results, including confidence intervals and effect sizes, and justify the choice of statistical tests. Apply and report corrections for multiple comparisons to control for Type I error.

6.    Clinical Relevance and Application:

      Flaw: The study does not establish clinically relevant cut-off values for pepsin concentration in BAL or discuss how these findings could be translated into clinical practice.

      Recommendation: Conduct further research to determine clinically relevant cut-off values for pepsin levels and explore the practical implications of using BAL pepsin concentration as a biomarker for EER in clinical settings.

7.    Consistency in Data Reporting:

      Flaw: The study uses inconsistent formats for reporting descriptive statistics (means ± SD vs. medians with IQR), which can cause confusion.

      Recommendation: Use a consistent format for reporting all descriptive statistics to enhance clarity and comparability of the data.

8.    Potential Bias in Patient Recruitment:

      Flaw: The study recruits patients who underwent bronchoscopy due to pulmonary abnormalities, which might introduce selection bias, as these patients might not represent the general population at risk for lung adenocarcinoma.

      Recommendation: Ensure that the study population is representative of the broader population at risk for lung adenocarcinoma by including a wider range of participants or using a randomized recruitment process.

9.    Novelty vs. Established Methods:

      Flaw: The use of BAL pepsin concentration as a biomarker for EER is novel but not yet an established or standardized method. This might limit the acceptance and reproducibility of the findings.

      Recommendation: Compare the BAL pepsin concentration method with established diagnostic techniques for EER to validate its reliability and applicability.

Identified Issues in the Study:

1.    Statistical Reporting Inconsistencies:

      Issue: The study reports p-values but lacks detailed statistical metrics such as confidence intervals and effect sizes. These metrics are essential for understanding the strength and reliability of the findings.

      Impact: Without these details, it is difficult to assess the true significance and clinical relevance of the results.

2.    Handling of Hemorrhagic BAL Samples:

      Issue: The rationale for excluding hemorrhagic BAL samples is not adequately explained. This exclusion could introduce bias or alter the interpretation of pepsin concentrations.

      Impact: The exclusion of these samples might skew the results and does not account for the potential significance of hemorrhagic samples in understanding pepsin levels.

3.    Potential Confounding Factors:

      Issue: The study controls for smoking status but does not address other potential confounding factors like diet, medication use, or coexisting respiratory conditions.

      Impact: Uncontrolled confounders can lead to inaccurate conclusions about the relationship between extraesophageal reflux and lung adenocarcinoma.

4.    Inadequate Explanation of Statistical Methods:

      Issue: The choice of statistical tests (e.g., Kruskal-Wallis and Wilcoxon tests) is not justified with an assessment of data normality. Also, there is no mention of corrections for multiple comparisons.

      Impact: The use of inappropriate statistical methods can lead to incorrect conclusions. Lack of multiple comparison correction increases the risk of Type I errors.

5.    Sample Size Limitations:

      Issue: The study includes a relatively small sample size, particularly for the sarcoidosis group (12 patients).

      Impact: Small sample sizes limit the power of the study to detect significant differences and reduce the generalizability of the findings.

6.    Lack of Mechanistic Insights:

      Issue: The study does not explore the biological mechanisms through which extraesophageal reflux might contribute to lung adenocarcinoma.

      Impact: Without mechanistic insights, it is challenging to understand the causal pathways and biological plausibility of the observed associations.

7.    Non-Standardized Method for Pepsin Detection:

      Issue: The use of pepsin concentration in BAL as a biomarker for EER is novel but not yet standardized or validated against established methods.

      Impact: This novelty, while interesting, limits the clinical applicability and acceptance of the findings without further validation.

8.    Potential Selection Bias:

     Issue: Patients were selected based on their need for bronchoscopy due to pulmonary abnormalities, which may not represent the general population.

      Impact: This selection criterion might introduce bias, as these patients could have other underlying conditions influencing the results.

Summary of Concerns

      Statistical Reporting: Lack of detailed statistical metrics and inadequate explanation of statistical methods.

      Sample Handling: Unclear rationale for excluding hemorrhagic BAL samples and potential bias introduced.

      Confounding Factors: Insufficient control of other potential confounding variables.

      Sample Size: Small sample size, particularly in the sarcoidosis group.

      Mechanistic Understanding: Absence of exploration into the biological mechanisms linking EER and lung adenocarcinoma.

      Method Validation: Use of a novel, non-standardized method for detecting pepsin in BAL without sufficient validation.

      Selection Bias: Potential bias due to the specific patient recruitment criteria.

Comments on the Quality of English Language

Author Response

Dear Reviewer,

Thank you for your comprehensive evaluation of our manuscript.

We acknowledge that the sample size, particularly in the sarcoidosis group, is small. This limitation may affect the statistical power and reliability of our findings. This is a pilot study and should be considered as offering preliminary findings that require further validation. We agree that future studies should aim for larger and more balanced sample sizes across all groups to enhance the validity and generalizability of the results

False positives in pepsin detection in hemorrhagic samples using ELISA can be caused by interference from haemoglobin and other blood proteins with the reagents used in the test. For this reason, hemorrhagic samples were excluded.

We recognize that diet and medication can influence reflux. Unfortunately, we do not have data on these factors. Although BMI can be measured, its relevance may be compromised due to the high prevalence of asthenic or cachectic conditions among our cancer patients cohort

The exact mechanism by which reflux may contribute to the development of lung adenocarcinoma is remains unclear. Our study did not focus on the causal pathways linking reflux to lung cancer tumorigenesis. Based on the research sequence of reflux-Barrett's esophagus-esophageal adenocarcinoma, we can hypothesize about the potential role of chronic inflammation induced by reflux, leading to increased proliferation and tumor transformation. Validating this theory though a single-cell transcriptomics study would be appropriate. This approach could provide valuable insights into how inflammation triggered by reflux influences the cellular and molecular processes involved in lung adenocarcinoma development. By examining the gene expression profiles of individual cells, we can better understand the heterogeneous responses to reflux and pinpoint the critical mechanisms underlying cancer development.

Thank you for the recommendations. The statistical analysis has been described in more detail, and diagnostic plots of the linear regression model have been added (Figures S1-S4 in the supplement), which show that there are no major violations of the linear model assumptions. Confidence intervals and effect sizes have also been included.

In the future, it will be essential to conduct further studies focused on establishing clinically relevant cut-off values for pepsin concentration in bronchoalveolar lavage (BAL). These studies should involve larger and more diverse cohorts, including healthy volunteers. Additionally, it will be crucial to identify and define all factors that influence pepsin levels in BAL

Currently, there is no gold standard for the diagnosis of extraesophageal reflux. Unlike established and standardized methods such as questionnaires, endoscopy with biopsy, pH-metry, and multichannel intraluminal impedance, the use of BAL pepsin concentration to detect extraesophageal reflux in the lower airways is not yet a standardized method. Unfortunately, while questionnaires like the Reflux Symptom Index (RSI) could be used in our patients, they may have significant overlap in symptoms with lung disease (e.g., cough, dyspnea, hoarseness, chest pain, etc.). The alternative methods mentioned would involve subjecting patients to additional invasive procedures. pH-metry is effective in detecting acid reflux but may not differentiate between acid and non-acid reflux, which can be clinically significant. Non-acid reflux, containing bile acids, pancreatic enzymes and pepsin, is particularly implicated in the pathogenesis of pulmonary diseases. Prolonged exposure to these substances can trigger inflammatory reactions, leading to airway remodelling and lung tissue damage.Multichannel intraluminal impedance-pH monitoring (MII-pH monitoring) offers the advantage of detecting both acid and non-acid reflux. However, it also presents challenges such as technical complexity, patient discomfort, and difficulties in interpretation.

We are aware that involving a broader population would contribute to the study's accuracy. Given that this is a pilot study, we considered the participation of healthy volunteers risky due to the invasiveness of bronchoscopic procedures.

Thank you once again for your valuable feedback. We appreciate your time and effort in reviewing our manuscript.

Best regards

Petra Zemanová

Round 2

Reviewer 3 Report

Comments and Suggestions for Authors

Dear Editor,

The authors found that their study did not support the hypothesis of a higher occurrence of extraesophageal reflux in non-smoking patients with lung adenocarcinoma. The number of cases is small. However, the revision shows significant improvements in the quality of the manuscript.

Sincerely

Reviewer 4 Report

Comments and Suggestions for Authors

-